# Environmental, Social, Governance & Financial Performance Disclosure for Large Firms: Is This Different for SME Firms?

Amir Gholami [ID], Peter A. Murray [ID] and John Sands *[ID]

Faculty of Business, Education, Law and Arts, University of Southern Queensland, Darling Heights, Toowoomba, QLD 4350, Australia; a.golame@gmail.com (A.G.); peter.murray@usq.edu.au (P.A.M.)
* Correspondence: john.sands@usq.edu.au

**Abstract:** This study examines the association between a firm's environmental, social and governance (ESG) performance and financial performance by examining the extent to which stakeholder and legitimacy theory help explain the effects on explanatory variables used in the study. Moreover, the study makes a novel contribution to existing ESG and performance-based studies by exploring the explanatory effects of ESG and firm performance over ten years. In addition, the study discusses the ESG-performance link of SMEs, thus advancing existing knowledge related to ESG in respect of SME performance. The study uses an extensive Australian sample from Bloomberg's database from 2007 to 2017, while panel regression analysis is applied to investigate the relationship between a firm's ESG performance and profitability. The robustness of the results is evaluated after incorporating several robustness checks to address methodological, endogeneity and causality issues related to a firm's ESG performance disclosure. The empirical findings of this study suggest that improving a firm's ESG performance is beneficial to all stakeholders of large firms in the long run but not for SME companies. The theoretical model suggests that listed SMEs do not disclose their ESG activities for various reasons, such as a lack of necessary resources. Specifically, the study extends scholarly understanding of existing theory and discusses the significance of the findings for future research.

**Keywords:** environmental; social and governance; firm financial performance; stakeholder theory; legitimacy theory; voluntary disclosure

## 1. Introduction

The irresponsible disclosure behaviour of firms leading up to the 2008–2009 period of financial turmoil has been a consistent global driver of corporate efforts to improve environmental, social and governance (ESG) performance [Global Economic Crisis (GFC)] [1,2]. Improving corporate ESG performance creates a win-win situation for firms, direct shareholders and stakeholders, and the overall economy [3,4]. This can be supported conceptually by both legitimacy and stakeholder theory, where the increasing number of social contacts between firms and their community of stakeholders is beneficial.

Firms are encouraged to enhance their ESG performance to achieve greater support from stakeholders [5–8] and provide improved financial performance for firms [9–11]. However, previous studies report inconclusive findings. For example, Whelan et al. [12] reviewed published or corporate studies between 2015 and 2020 that concentrated on the association between ESG and a firm's financial performance, such as ROA and ROE. Their analysis identified positive, neutral or negative results of prior studies, which is consistent with the results produced by Margolis and Walsh [13]. Margolis and Walsh [13] found only a positive association. However, these studies tested the association between ESG and financial performance for large firms with largely diverse ownership. By comparison, smaller firms, such as small to medium enterprises (SMEs), tend to have narrower and more concentrated ownership structures, with many SMEs not practising the same level of disclosure (e.g., Zadeh and Eskandari [14]). Al Fadli et al. [15] argue that this difference

in disclosure levels may be due to narrow ownership of smaller firms providing access to more proprietary information than experienced by large firms' shareholders and internal stakeholders, and therefore, the same level of disclosure is not considered necessary.

The favourable association between ESG and performance is supported by other studies [16–19]. Additionally, the lack of consensus caused by inconclusive findings of prior literature on the relationship has left this line of study unresolved, thus prompting new research questions [16,20,21]. Moreover, similar to large firms but for different reasons, the link between ESG and SME performance is not yet explored, while the few studies that exist have tended to examine SME sustainability postures and performance [22]. Linking SME ESG to SME performance is important because we contend that the social contract between firms and their stakeholders [23] will be equally important for these firms. Improving an SME value system might also be congruent with the value system of societal expectations [24], among other things.

Australia is a market-based economy, similar to many countries, such as the US, UK, New Zealand, Singapore, Germany, Ireland, and Canada, where larger listed companies have a diverse share ownership base. In contrast to this diverse share ownership base, the listed SME companies have a narrower share ownership base, which is similar to a number of Nordic and Germanic-based European countries identified, initially, by Hofstede [25] and adopted by Gray [26], such as the Netherlands, Austria, Switzerland, and France. This broad representation of different ownership bases in Australia provides a good platform for research that may be affected by the disclosure practices associated with these characteristics of ownership variation.

Lys et al. [27] argue that the financial benefits from improving corporate ESG performance disclosure emerge in subsequent periods, representing a lag in receiving financial performance benefits. This argument is supported by the model simulation findings by Bianchi et al. [28]. Therefore, prior studies may neglect a key variable by failing to consider time and temporal lags in their investigation. To overcome these failures, panel data analysis was selected for the current study because of its superior attributes over pure time series data analysis since it (1) provides a model of the common and individual behaviours of (companies) groups, (2) detects and measures statistical effects that cannot be detected by pure time series, and (3) minimises any estimation biases that may arise from aggregating groups into a single time series. A critical difference between time series and panel data is that time series focus on a single individual at multiple time intervals while panel data focus on multiple individuals at multiple time intervals. This study performs several tests, including robustness and sensitivity checks, to address the issues regarding differences in industries [29] or endogeneity issues [30], while panel regression analysis, including the firm's size effects, is used to negate any concern about unobserved firm-specific variables or missing elements.

This study is expected to contribute to the ESG-performance disclosure literature related to SMEs and enable an enhanced understanding of the economic consequences of a firm's ESG performance, particularly for large firms over time. The use of panel data to address the ESG-performance lag is expected to enhance scholarly understanding of corporate ESG performance significantly. Investors are increasingly taking corporate ESG performance disclosure into account when making investment decisions. By promoting ESG performance, large firms may capture new customers, and through these new customers, there should be an increase in cash inflows from increased sales (or increased deposits for financial institutions), which eventually impacts firms' financial performance. Similarly, SMEs and their stakeholders will be better informed about matching ESG requirements to performance as a result of this study. Taken together, this study explores these gaps and explicitly addresses the time issues by asking the following questions:

Research question (RQ1): What effect does ESG performance have on large and SME firms' financial performance?

Research question (RQ2): What effect does ESG performance have on the financial performance for large firms?

Research question (RQ3): What effect does ESG performance disclosure have on the financial performance for SME firms?

Under the following sections: Section 2 presents the theoretical discussion, literature review and hypotheses development, exploring the research questions. Section 3 discusses the sample data and research method used in the gathering of data for the analysis. Section 4 summarizes the results. Section 5 outlines the results of the robustness tests, followed by the conclusion section in Section 6, which includes the implications for practice as well as the limitations and implications for future research.

## 2. Literature Review, Theoretical Framework and Hypotheses

The concept of corporate ESG performance disclosure has been discussed extensively in the academic literature over the past four decades [16,31]. The focus of firms' goals has extended from solely addressing shareholders' needs to include all stakeholders' needs. This extended focus is based on the recognition that there are diverse groups of stakeholders, such as local communities, employees, and customers, who care about corporate ESG performance disclosure [32].

### 2.1. ESG and Firm Performance

Social contract theory provides a basis for the theoretical social contract between firms and their stakeholders [23,33]. There are two aspects to this theoretical social contract. The first aspect is related to investors who wish to maximise their investment return while the second relates to the broader issues of operational sustainability and corporate governance [34]. These two aspects form the current norms and expectations of the broader community. That is, firms' value system is perceived as being congruent with the value system of the larger social system of which the entity is a part and must conform to the terms of the social contract [35]. This conformity with social contracts consists of sets of formal and informal agreements that apply to the firms themselves and their various stakeholders, that is, social contracts comprising sets of formal and informal agreements between societal groups and obligations toward each other [36]. For example, firms' theoretical social contract with the broader community includes banks and other large fund providers, institutional investors, employees, as well as customers [24,37]. These external and internal stakeholders also have a theoretical social contract with the broader community and are motivated to project their conformity with both aspects of the social contract. The fundamental association between parties to this social contract is a reciprocation legitimacy identified in legitimacy theory [33]. Here, the managerial branch of stakeholder theory [5] explains the motivation of managers to meet the ESG disclosure needs of dominant stakeholder groups that affect firms' ongoing survival.

The managerial branch of stakeholder theory explains how managing the relationship with different stakeholder groups through corporate disclosure helps to improve firm financial performance [5,6]. Different groups' needs are prioritised through mapping the salience of each group based on three criteria: power, authority, and urgency [38]. The highly salient stakeholders' group is categorised as having a high rating for all three criteria. Therefore, firms are motivated to enhance ESG performance disclosure to improve reputation and accountability [39], which, according to scholars, leads to increased financial performance [9,10,40].

Although the most prevalent view of the relationship between ESG and firm financial performance is positive, there are some contrasting findings in the literature [41–43]. For instance, although the relationship is reported to be moderately positive in some literature [16,20,41], some report neither a significant relationship [44] nor a neutral one [45]. Environmental performance has been positively related to return on assets (ROA) [46,47], which is consistent with some meta-analysis studies [48–50].

From the stakeholder theory viewpoint, prior literature [6,51,52], and the mode posited recently by Fatemi et al. [53] responsible behaviour of a firm positively impacts a firm's financial performance. It can also mitigate potential damage to a firm's market value through

a commitment to socially responsible behaviour [54]. Paredes-Gazquez et al. [55] reveal a positive association between corporate social engagement and financial performance, while numerous studies find a positive association between corporate governance and financial performance [56,57]. Corporate governance is the main driver of firm sustainability, and ESG investment is critical [58]. Rabaya and Saleh [59] found that ESG disclosure had a positive influence on the competitive advantage at the firm level over ten years, which led to increases in financial performance. For instance, within the strategy field, Lavie, et al. [60] suggest that 'over time, repeated use of exploitation routines generates reliable feedback that enables organisations to refine their existing competencies further and evaluate better the likely success of exploitation efforts'. Regarding ESG and financial performance outcomes for SMEs, it is thus less likely that financial performance will be influenced over a shorter period after implementing ESG innovations. More likely, it takes longer periods to embed ESG capabilities such that there is a lag in performance. Also, for ESG studies thus far, a longitudinal research design with some exceptions (e.g., Drempetic, Klein and Zwergel [22]) has not been reported in prior research within an SME context.

This study explores the association between aggregated and disaggregated elements of a firm's ESG performance and large firms' profitability. Thus, the first hypothesis is:

**Hypothesis 1 (H1).** *There is a positive relationship between a firm's ESG performance and profitability.*

### 2.2. Firm Characteristics and Performance

Another stream of the literature delves into the diverse components of ESG performance and its impact on financial performance. Cormier and Magnan [61] propose that the trade-off in relation to corporate ESG performance disclosure may vary, particularly in respect of large and SME firm characteristics [22,62]. Therefore, a firm's characteristics, such as its size, management structure, reputation and media exposure, should be considered [63].

Friedman et al. [64] provide the most dissenting viewpoint, arguing that corporate ESG performance disclosure imposes costs higher than its benefits and causes a misallocation of corporate resources. Nevertheless, the literature focuses on the modest positive relationship between a firm's ESG performance and profitability [41,42,65–67]. However, higher costs through implementing ESG initiatives and increasing ESG performance disclosure may be more achievable by larger firms. Extant research identifies the higher cost of implementing ESG [68] plus the higher cost of increasing the level of ESG performance disclosure [69]. Barauskaite and Streimikiene [21] conclude that ESG initiatives lead to both additional benefits and additional costs for businesses.

Some studies recommend that good ESG performance can enhance a firm's reputation and operational performance [10,70]. Cornett et al. [71] argue that larger financial firms follow sustainable behaviours on a larger scale than smaller ones. This trend may be present for larger and smaller firms irrespective of industry. Moreover, a recent study by Drempetic, Klein and Zwergel [22] found that ESG scores do not realistically measure the sustainability performance of a company and that firm size—such as corporate firms—are more advantaged as they have access to data availability and the necessary resources for providing ESG data. Interestingly, these scholars found that 'ESG scores [are] distorted in favour of large companies because ESG scores are dependent on resources for providing ESG data. In respect of the current study's aims, more needs to be known about the influence of ESG on the financial performance of SMEs since these limited studies have mainly focused on ESG and sustainability as distinct from ESG and financial performance.

Rabaya and Saleh [59] dissected their analysis into large and small firms and found that although there is a percentage difference between the two firm sizes, both small and large firms experienced an increase between ESG implementation and firm performance. The efforts implemented to address ESG issues constitute a cost to a firm that may lead to lower profitability, but cost-effective improvements from reduced waste and energy-saving benefits tend to offset the initial costs [72]. In summary, large firms have the financial capacity to pursue sustainable business on a greater scale than smaller firms, and therefore,

the large firms' greater investment in ESG performance disclosure may result in these firms having a greater competitive advantage and subsequent profitability.

**Hypothesis 2 (H2).** *The relationship between a firm's ESG performance and its financial performance differs for large firms compared to SME firms.*

### 3. Data and Methodology

*3.1. Environmental, Social and Governance (ESG)*

For this study, the firms' mostly voluntary ESG performance disclosure (ESG) is the principal predictor variable. The ESG disclosure score is used as a measure of a firm's ESG performance. The high demand for ESG-related information is reflected in several databases, such as Assets4, Bloomberg and RepRisk, providing ESG disclosure scores. Never-theless, inconsistencies exist in the scores awarded by these databases [73,74]. Bloomberg's measurement for scoring a firm's ESG performance is regarded as the most consistent metric among these databases [73]. Bloomberg's measuring scale is based on 120 indicators, including diverse environmental, social, and governance performance elements. This scaling system has been used in numerous academic papers [75,76]. Following Di Giuli and Kostovetsky [77], this study normalises the ESG scores to have a notional standard scale.

*3.2. Sample and Data*

Aggregated ESG performance data and disaggregated environmental (ENV), social (SOC) and governance (GOV) elements were gathered for all listed Australian firms from the Bloomberg database for the period 2007–2017. This period of data selection is uniform with the period used in past studies reflecting the model simulation period (e.g., Lys, Naughton and Wang [27], Bianchi, Cosenz and Marinković [28]). The reason why 2007 was the commencement date for data used in this study is that firms have mostly engaged in ESG disclosure activities since 2007 due to its availability and because of the GFC between 2008–2009 prompted firms to improve their ESG disclosure. Any firms that have not disclosed one of the ESG elements were not included in the analysis. A firm's return on assets (ROA) is the main proxy for financial performance. Other firms' specific financial data are used in the analysis. These data include total assets (LNTA) as a proxy for the size of the firm; property, plant and equipment (PPE); the firm's capital expenditure (capex); total revenue (growth); cash ratio (cash); and debt ratio (leverage). The details are provided in the attached Appendix A.

More than 125,000 items of data from Australian publicly listed firms between 2007 and 2017 were included in the initial sample. We matched the collected observations with the firm's ESG performance disclosure score after obtaining data on financial performance and other financial data. The data for the key variables were completed after excluding the missing data, which resulted in a total sample of 31,115 observations for 3422 publicly listed firms. Tables 1–3 show the firm sample composition by year, industry, and the descriptive statistics for each variable across all industries and firms, irrespective of firm size.

Descriptive statistics for this study are provided in Table 3. Variables are winsorized at a scale of 1% to 99% to control the influence of outliers. The ESG score variances provided in this table support the representative nature of the scores and the appropriateness of evaluating the impact of a firm's ESG performance on its financial performance.

Our methodology includes dividing the final sample into two subsamples of large and SME firms as a test for H2 as hypothesised. Zadeh and Eskandari [14] found that prior studies have separately used six different measures of firm size to test the association between firm size and level of disclosure. Shalit and Sankar [78] found that net total assets (NTA) may be interchangeable with equity as a measure of size. Therefore, for this longitudinal study for the period from 2007 to 2017, firm size was measured using the natural log of total assets (LNTA) based on Bloomberg's data. Table 4 dissects the

descriptive statistics for each variable into large (Panel A) and small to medium (SME) firms (Panel B), using the mean of LNTA as the dissection point.

**Table 1.** Composition of each sample year by Firm Size.

| Year | Large Firms | SME Firms |
|---|---|---|
| 2007 | 98 | 146 |
| 2008 | 106 | 161 |
| 2009 | 109 | 163 |
| 2010 | 116 | 167 |
| 2011 | 135 | 160 |
| 2012 | 145 | 159 |
| 2013 | 149 | 165 |
| 2014 | 159 | 181 |
| 2015 | 170 | 194 |
| 2016 | 186 | 184 |
| 2017 | 194 | 175 |
| Total | 1567 | 1855 |

This table represents the yearly composition of firms in the sample.

**Table 2.** Composition of each sample industry.

| Year | Observations | % |
|---|---|---|
| Basic Materials | 754 | 22% |
| Consumer Non- Cyclical | 595 | 17% |
| Financial | 575 | 17% |
| Energy | 386 | 11% |
| Consumer Cyclical | 377 | 11% |
| Industrial | 351 | 10% |
| Communications | 201 | 6% |
| Technology | 108 | 3% |
| Utilities | 57 | 2% |
| Diversified | 18 | 1% |
| Total | 3422 | 100% |

This table represents the industry composition of firms in the sample.

**Table 3.** Descriptive statistics for variables.

| | No. | Mean | Std | P25 | Median | P75 |
|---|---|---|---|---|---|---|
| ESG | 3422 | 2.9293 | 0.4711 | 2.5413 | 2.8639 | 3.1939 |
| ENV | 1315 | 2.451 | 1.015 | 1.537 | 2.579 | 3.329 |
| SOC | 2020 | 3.062 | 0.68 | 2.759 | 3.201 | 3.507 |
| GOV | 2411 | 3.878 | 0.159 | 3.758 | 3.876 | 3.947 |
| ROA | 2422 | 1.9222 | 0.9488 | 1.4338 | 2.0073 | 2.5129 |
| LNTA | 3422 | 6.4935 | 2.147 | 5.0982 | 6.2964 | 7.7873 |
| PPE | 3422 | 0.6598 | 1.0661 | 0.0419 | 0.2241 | 0.8047 |
| Capex | 3422 | 0.224 | 0.6353 | 0.0101 | 0.0396 | 0.1545 |
| Growth | 3422 | 0.1074 | 0.6438 | -0.005 | 0.0443 | 0.1779 |
| Cash | 3422 | 0.124 | 0.1424 | 0.0249 | 0.0658 | 0.1686 |
| Leverage | 3422 | 0.4286 | 0.2613 | 0.2553 | 0.4237 | 0.5757 |

This table presents the descriptive statistics for all variables, including all firms. ESG, ENV, SOC, and GOV reflect firm aggregated ESG, environmental, social, and governance scores.

**Table 4.** Panel A—statistics for variables of SME firms. Panel B—statistics for variables of Large firms.

| Panel A | | | | | | |
|---|---|---|---|---|---|---|
| | **No.** | **Mean** | **Std** | **P25** | **Median** | **P75** |
| ESG | 1855 | 2.6697 | 0.3052 | 2.4275 | 2.6541 | 2.8871 |
| ENV | 295 | 1.5985 | 0.7643 | 0.8439 | 1.5371 | 2.0479 |
| SOC | 746 | 2.7474 | 0.6546 | 2.3538 | 2.9600 | 3.2011 |
| GOV | 1039 | 3.8001 | 0.1206 | 3.7578 | 3.7986 | 3.8756 |
| ROA | 1093 | 2.2181 | 0.9567 | 1.8089 | 2.3391 | 2.8215 |
| LNTA | 1855 | 4.9649 | 1.1852 | 4.3276 | 5.2202 | 5.8727 |
| PPE | 1855 | 0.6459 | 1.2431 | 0.0189 | 0.1391 | 0.6470 |
| Capex | 1855 | 0.2915 | 0.8188 | 0.0042 | 0.0288 | 0.1587 |
| Growth | 1855 | 0.1221 | 0.8182 | 0.0122 | 0.0256 | 0.2057 |
| Cash | 1855 | 0.1757 | 0.1658 | 0.0438 | 0.1163 | 0.2708 |
| Leverage | 1855 | 0.3418 | 0.2639 | 0.1419 | 0.3201 | 0.4899 |
| **Panel B** | | | | | | |
| | **No.** | **Mean** | **Std** | **P25** | **Median** | **P75** |
| ESG | 1567 | 3.2147 | 0.4620 | 2.8638 | 3.1666 | 3.5983 |
| ENV | 1020 | 2.6975 | 0.9423 | 2.0479 | 2.9234 | 3.4975 |
| SOC | 1274 | 3.2462 | 0.6255 | 2.9600 | 3.3347 | 3.6531 |
| GOV | 1372 | 3.9367 | 0.1589 | 3.8379 | 3.8938 | 4.0455 |
| ROA | 1329 | 1.6787 | 0.8700 | 1.2117 | 1.7955 | 2.2577 |
| LNTA | 1567 | 8.3029 | 1.5372 | 7.1187 | 7.9511 | 9.0264 |
| PPE | 1567 | 0.6762 | 0.8080 | 0.0855 | 0.3523 | 0.9343 |
| Capex | 1567 | 0.1441 | 0.2755 | 0.0169 | 0.0515 | 0.1487 |
| Growth | 1567 | 0.0901 | 0.3351 | −0.0181 | 0.0521 | 0.1516 |
| Cash | 1567 | 0.0627 | 0.0694 | 0.0169 | 0.0384 | 0.0831 |
| Leverage | 1567 | 0.5312 | 0.2173 | 0.3863 | 0.5023 | 0.6658 |

This table presents the descriptive statistics for the variables for the SME and large firms.

Table 4 presents the summary descriptive statistics for large and SME firms. Established statistics provided in Table 4 show an average ESG score range from 2.67 for SME firms to 3.21 for large firms. Additionally, variations ranged from 2.43 for SME firms to 2.86 for large firms at the 25th percentile, and 2.89 and 3.60, respectively, at the 75th percentile. Therefore, the statistics show sufficient variation in the ESG disclosure score between firms' sizes to examine the impact of firms' sizes on the association between corporate ESG performance and financial profitability.

This study utilises an accounting measure of return on assets (ROA) as a dependent variable, which Brooks and Oikonomou (2018) [20] recommended as the most accurate measure of accounting performance. ROA has been used in several sustainability studies to measure financial performance [79–81]. Therefore, this study follows Soana [81] and uses ROA to compare financial performance for large and SME firms.

### 3.3. Estimation Models

A panel regression analysis was applied to sample firms to examine the first hypothesis and provide responses to the respective research questions. Consistent with H1, we propose that a firm's ESG performance is positively related to profitability. Here, ROA is used as a proxy for firm profitability. The ROA is not a market-sensitive metric and is traditionally used for profitability comparisons between SME and large firms. The regression of ROA over a firm's ESG performance and its other disaggregated elements result in the comprehension of the potential economic impacts of ESG performance on a firm's profitability.

$$\text{ROA}_{i,t} = \beta_0 + \beta_1 \text{ESG}_{i,t} + \beta_3 \text{LNTA}_{i,t} + \beta_2 \text{PPE}_{i,t} + \beta_4 \text{CAPEX}_{i,t} + \beta_6 \text{GROWTH}_{i,t} + \beta_7 \text{CASH}_{i,t}$$
$$+ \beta_5 \text{LEVERAGE}_{i,t} + \text{IndustryFixedEffect}_t + \text{YearFixedEffect}_t + \varepsilon_{it}$$

(1)

Also, other characteristics of firms that impact operational performance and are used by Aggarwal et al. [82] are included in the analysis. In particular, the following characteristics are examined: the size of a firm, which is operationalized using the log of its total assets (LNTA); the property, plant, and equipment (PPE) using the PPE ratio; leverage based on the debt ratio with total liabilities as the numerator and total assets as the denominator; capital expenditure (capex) calculated through a firm's capital expenditure as the numerator and its total revenue as the denominator; a firm's growth represented by the percentage change in a firm's revenue between periods; and cash available denoted by the cash ratio, consisting of cash items in the balance sheet as the numerator and total assets as the denominator.

The LNTA in model 2 differs from that in model 1, where LNTA was the log of its total assets. In model 2, LNTA was employed in model 2 as a dichotomy of LNTA, based on the mean of LNTA, that provided large or SME-sized firms (large firms = 1 and SME firms = 0). This dissection of the firms into large and SME firms permitted the analysis to text Hypothesis 2. The findings from the panel analysis to appraise the second hypothesis are discussed in the following section.

$$
\begin{aligned}
\text{ROA}_{i,t} = \ & \beta_0 + \beta_1 \text{ESG}_{i,t} + \beta_3 \text{LNTA}_{i,t} + \beta_2 \text{PPE}_{i,t} + \beta_4 \text{CAPEX}_{i,t} + \beta_6 \text{GROWTH}_{i,t} + \beta_7 \text{CASH}_{i,t} \\
& + \beta_5 \text{LEVERAGE}_{i,t} + \text{IndustryFixedEffect}_t + \text{YearFixedEffect}_t + \varepsilon_{it}
\end{aligned}
\tag{2}
$$

## 4. Results

### 4.1. Main Regression Results

Regression analyses were conducted to test the two hypotheses as well as additional analyses to identify the impact of each of the three components of the overall ESG score individually on large and SME firms' financial performance. Tables 5–10 provide affirmative comments that confirm that both industry fixed effect and year fixed effect in the analysis controlled for any industry or time-variant attributes' impact or bias on the independent or dependent variables of this panel data analysis [83].

**Table 5.** Main regression results—aggregated ESG on ROA.

| Variables | ROA |
|---|---|
| ESG | 0.2625 *** |
| | (0.0523) |
| LNTA | −0.1659 *** |
| | (0.0141) |
| PPE | −0.0589 * |
| | (0.0323) |
| CAPEX | 0.05823 |
| | (0.0715) |
| GROWTH | 0.1528 *** |
| | (0.0365) |
| CASH | 0.5331 *** |
| | (0.1721) |
| LEVERAGE | −0.4063 *** |
| | (0.0981) |
| Const | 2.4430 *** |
| | (0.1254) |
| Year FE | Yes |
| Industry FE | Yes |
| No of Obs | 3422 |
| Adj R-sq | 0.1625 |

This table shows the main panel regression result of firms' profitability on their ESG score. Superscripts *** and * indicate significance at 1% and 10% levels, respectively.

**Table 6.** Baseline Large Versus SME Firm comparison analysis.

| Variables | Model (1) | Model (2) |
|---|---|---|
| ESG | 0.3778 *** | 0.1165 |
| | (0.0583) | (0.1017) |
| LNTA | −0.1738 *** | −0.0075 |
| | (0.0189) | (0.0375) |
| PPE | −0.0264 | −0.1558 *** |
| | (0.0385) | (0.0591) |
| CAPEX | 0.0909 | −0.1359 |
| | (0.1233) | (0.1021) |
| GROWTH | 0.1744 *** | 0.1526 *** |
| | (0.0652) | (0.0460) |
| CASH | 2.0574 *** | 0.5725 ** |
| | (0.3168) | (0.2293) |
| LEVERAGE | −0.7269 *** | 0.0687 |
| | (0.1191) | (0.1592) |
| Const | 2.1719 *** | 1.8489 *** |
| | (0.1592) | (0.3243) |
| Year FE | Yes | Yes |
| Industry FE | Yes | Yes |
| No of Obs | 3422 | 3422 |
| Adj R-sq | 0.2156 | 0.2181 |

This table shows the main panel regression result of a firm's profitability on their ESG score. The first Columns (Model 1) report the panel regression result for large firms, and the second columns (Model 2) report the result for the SME firms. Superscripts *** and ** indicate statistical significance at 1% and 5% levels, respectively.

**Table 7.** Disaggregated ESG analysis—All companies.

| Variables | ROA | | | |
|---|---|---|---|---|
| | (1) | (2) | (3) | (4) |
| ESG | 0.2625 *** | | | |
| | (0.0523) | | | |
| ENV | | 0.1014 *** | | |
| | | (0.0302) | | |
| SOC | | | 0.1211 *** | |
| | | | (0.0376) | |
| GOV | | | | 0.6420 *** |
| | | | | (0.1463) |
| LNTA | −0.1659 *** | −0.1690 *** | −0.1560 *** | −0.1655 *** |
| | (0.0141) | (0.1906) | (0.0145) | (0.0143) |
| PPE | −0.0589 * | −0.0247 | −0.0393 | −0.0358 |
| | (0.0323) | (0. 0486) | (0.0404) | (0.0365) |
| Capex | 0.05823 | 0.0355 | 0.0319 | 0.0176 |
| | (0.0715) | (0.1619) | (0.1119) | (0.0984) |
| Growth | 0.1528 *** | 0.4078 *** | 0.1837 *** | 0.15349 *** |
| | (0.0365) | (0.0840) | (0.0470) | (0.0433) |
| Cash | 0.5331 *** | 1.5740 *** | 1.1654 *** | 0.9608 *** |
| | (0.1721) | (0.3159) | (0.2183) | (0.1983) |
| Leverage | −0.4063 *** | −0.5652 *** | −0.5372 *** | −0.5447 *** |
| | (0.0981) | (0.1384) | (0.1133) | (0.1080) |
| Const | 2.4430 *** | 3.0888 *** | 2.8621 *** | 0.8080 |
| | (0.1254) | (0.1214) | (0.1186) | (0.5236) |
| Year FE | Yes | Yes | Yes | Yes |
| Industry FE | Yes | Yes | Yes | Yes |
| No of Obs | 2422 | 1097 | 1641 | 1918 |
| Adj R-sq | 0.1625 | 0.2557 | 0.2331 | 0.2206 |

This table shows the findings of firms' ROA on not only aggregated ESG performance but also disaggregated ESG elements: environmental (ENV), social (SOC) and governance (GOV). Superscript asterisks *** and * represent 1% and 10% levels of significance, respectively.

**Table 8.** Disaggregated ESG analysis—large companies.

| Variables | ROA | | | |
| --- | --- | --- | --- | --- |
| | **(1)** | **(2)** | **(3)** | **(4)** |
| ESG | 0.3778 *** | | | |
| | (0.0583) | | | |
| ENV | | 0.1189 *** | | |
| | | (0.0320) | | |
| SOC | | | 0.1633 *** | |
| | | | (0.0417) | |
| GOV | | | | 0.8853 *** |
| | | | | (0.1675) |
| LNTA | −0.1738 *** | −0.1525 *** | −0.1353 *** | −0.1554 *** |
| | (0.0189) | (0.0213) | (0.1765) | (0.0188) |
| PPE | −0.0264 | −0.0040 | −0.0017 | 0.0007 |
| | (0.0385) | (0.0500) | (0.0454) | (0.0408) |
| Capex | 0.0909 | 0.0278 | 0.1118 | 0.0113 |
| | (0.1233) | (0.1687) | (0.1571) | (0.1320) |
| Growth | 0.1744 *** | 0.3718 *** | 0.2047 *** | 0.1714 ** |
| | (0.0652) | (0.1140) | (0.0789) | (0.0696) |
| Cash | 2.0574 *** | 2.4600 *** | 2.3738 *** | 2.3902 *** |
| | (0.3168) | (0.4088) | (0.3465) | (0.3366) |
| Leverage | −0.7269 *** | −0.7528 *** | −0.6734 *** | −0.7069 *** |
| | (0.1191) | (0.1454) | (0.1287) | (0.1253) |
| Const | 2.1719 *** | 2.9259 *** | 2.4969 *** | −0.2965 |
| | (0.1592) | (0.1473) | (0.1540) | (0.5947) |
| Year FE | Yes | Yes | Yes | Yes |
| Industry FE | Yes | Yes | Yes | Yes |
| No of Obs | 2422 | 1097 | 1641 | 1918 |
| Adj R-sq | 0.2156 | 0.2155 | 0.2073 | 0.2116 |

This table shows the findings of large firms' ROA on not only aggregated ESG performance but also disaggregated ESG elements: environmental (ENV), social (SOC) and gofvvernance (GOV). Superscript asterisks *** and ** represent 1% and 5% levels of significance, respectively.

**Table 9.** Disaggregated ESG analysis—SME companies.

| Variables | ROA | | | |
| --- | --- | --- | --- | --- |
| | **(1)** | **(2)** | **(3)** | **(4)** |
| ESG | 0.1165 | | | |
| | (0.1017) | | | |
| ENV | | −0.0713 | | |
| | | (0.0822) | | |
| SOC | | | −0.1266 | |
| | | | (0.0818) | |
| GOV | | | | 0.2469 |
| | | | | (0.2794) |
| LNTA | −0.0075 | 0.2726 ** | 0.2531 | 0.0622 |
| | (0.0375) | (0.1906) | (0.0647) | (0.0511) |
| PPE | −0.1558 *** | −0.2506 | −0.2997 | −0.2290 *** |
| | (0.0591) | (0.1801) | (0.1023) | (0.0821) |
| Capex | −0.1359 | −0.1593 | 0.2551 | 0.0908 |
| | (0.1021) | (0.4818) | (0.1909) | (0.1605) |
| Growth | 0.1526 *** | 0.4859 *** | 0.1938 *** | 0.1328 ** |
| | (0.0460) | (0.1391) | (0.0619) | (0.0568) |
| Cash | 0.5725 ** | 0.5115 | 0.6349 ** | 0.6470 ** |
| | (0.2293) | (0.5562) | (0.3138) | (0.2662) |
| Leverage | 0.0687 | 0.1393 | −0.2175 | −0.2548 |
| | (0.1592) | (0.3902) | (0.2201) | (0.1940) |
| Const | 1.8489 *** | 0.7876 | 2.270 *** | 1.0903 |
| | (0.3243) | (0.7328) | (0.3908) | (1.0676) |
| Year FE | Yes | Yes | Yes | Yes |
| Industry FE | Yes | Yes | Yes | Yes |
| No of Obs | 2422 | 1097 | 1641 | 1918 |
| Adj R-sq | 0.2181 | 0.1056 | 0.0608 | 0.0459 |

This table shows the findings of SME firms' ROA on not only aggregated ESG performance but also disaggregated ESG elements: environmental (ENV), social (SOC) and governance (GOV). Superscript asterisks *** and ** represent 1% and 5% levels of significance, respectively.

**Table 10.** Robustness analysis—All companies.

| Variables | ROA | | | |
| --- | --- | --- | --- | --- |
| | **(1)** | **(2)** | **(3)** | **(4)** |
| ESG | 0.2325 *** | | | |
| | (0.0523) | | | |
| ENV | | 0.1014 *** | | |
| | | (0.0302) | | |
| SOC | | | 0.0611 | |
| | | | (0.0376) | |
| GOV | | | | 0.6420 *** |
| | | | | (0.1463) |
| LNTA | −0.1659 *** | −0.1690 *** | −0.1560 *** | −0.1656 *** |
| | (0.0141) | (0.0191) | (0.0135) | (0.0123) |
| PPE | −0.0569 * | −0.0247 | −0.0393 | −0.0358 |
| | (0.0313) | (0.0476) | (0.0401) | (0.0365) |
| Capex | 0.0582 | 0.0355 | 0.0319 | 0.0176 |
| | (0.0705) | (0.1619) | (0.1119) | (0.0984) |
| Growth | 0.1582 *** | 0.4078 *** | 0.1837 *** | 0.1349 *** |
| | (0.0361) | (0.0838) | (0.0468) | (0.0433) |
| Cash | 0.5331 *** | 1.5417 *** | 1.1654 *** | 0.9608 *** |
| | (0.1721) | (0.3159) | (0.2183) | (0.1983) |
| Leverage | −0.4063 *** | −0.5652 *** | −0.5373 *** | −0.5447 *** |
| | (0.0981) | (0.1374) | (0.1133) | (0.1080) |
| Const | 2.4429 *** | 3.0888 *** | 2.8632 *** | 0.8080 |
| | (0.1254) | (0.1214) | (0.1185) | (0.5236) |
| Year FE | Yes | Yes | Yes | Yes |
| Industry FE | Yes | Yes | Yes | Yes |
| No of Obs | 2422 | 1097 | 1641 | 1918 |
| Adj R-sq | 0.1620 | 0.2557 | 0.2231 | 0.2206 |

This table provides the results of examining the robustness of the estimation models. Superscript asterisks *** and * indicate significance at the 1% and 10% levels, respectively.

### 4.1.1. Environmental, Social and Governance (ESG) Performance and Profitability

We use the ROA ratio as a proxy for firm profitability and to test our first estimation models. The results are presented in Table 5.

The firm's ESG performance coefficient of 0.2625 is significant at the 1% level statically (*t*-statistic = 5.02, standard error = 0.0523). These statistics support hypothesis (H1) that higher ESG performance is associated with higher profitability. The results show that a one-standard-deviation increase in the firm ESG performance leads to a 1.37% increase in ROA (0.0523 × 0.2625). Therefore, we conclude that a firm with higher ESG performance performs better financially over the full study period. These findings are consistent with recent literature [41,84–86]. The findings for other variables are also in line with the current findings in the literature, and their coefficients follow the same direction. In line with the findings of Aggarwal, Erel, Stulz and Williamson [83], negative correlations are found between ROA and LNTA and leverage. Furthermore, a positive correlation is found between ROA and the liquidity ratio (cash), representing the ratio of cash to total assets, which is consistent with Konijn et al. [87]. In line with the prediction by King and Santor [88], a positive correlation is found between ROA and revenue growth (growth) over time using panel data regression. Therefore, the first hypothesis of this study is supported.

### 4.1.2. Firm Size Analysis

Table 6 shows the firms' size comparison results for large firms (column a) and SME firms (column b) in testing H2 as hypothesized. The findings of column (a) partially support the association implied by H2 that large firms with higher ESG performance disclosure achieve higher profitability. Column (b) shows no significant findings, therefore, no support for the proposed association between higher ESG performance disclosure by SME firms

and their achieving higher profitability, which supports the implicit difference between large and SME firms in H2. That is, a one standard deviation increase in ESG performance disclosure for large firms results in a 2.20% increase ($0.0583 \times 0.3778$) in profitability compared to no significant increase in profitability (ROA) for SME firms. Consequently, there is partial support for H2.

### 4.1.3. Additional Analyses

Additional follow-up analyses are used to identify which of the three ESG elements impact all firms' financial performance, large firms' financial performance, and SME firms' financial performance. The disaggregation of the three elements of firms' ESG scores assists in comprehending the three dimensions of ESG performance (ENV, SOC or GOV) has a significant association with all, large, and SME firms' profitability:

$$
\begin{aligned}
\text{ROA}_{i,t} = \ & \beta_0 + \beta_1 \text{ENV}_{i,t} + \beta_3 \text{LNTA}_{i,t} + \beta_2 \text{PPE}_{i,t} + \beta_4 \text{CAPEX}_{i,t} + \beta_6 \text{GROWTH}_{i,t} + \beta_7 \text{CASH}_{i,t} \\
& + \beta_5 \text{LEVERAGE}_{i,t} + \text{IndustryFixedEffect}_t + \text{YearFixedEffect}_t + \varepsilon_{it}
\end{aligned}
\tag{3}
$$

$$
\begin{aligned}
\text{ROA}_{i,t} = \ & \beta_0 + \beta_1 \text{SOC}_{i,t} + \beta_3 \text{LNTA}_{i,t} + \beta_2 \text{PPE}_{i,t} + \beta_4 \text{CAPEX}_{i,t} + \beta_6 \text{GROWTH}_{i,t} + \beta_7 \text{CASH}_{i,t} \\
& + \beta_5 \text{LEVERAGE}_{i,t} + \text{IndustryFixedEffect}_t + \text{YearFixedEffect}_t + \varepsilon_{it}
\end{aligned}
\tag{4}
$$

$$
\begin{aligned}
\text{ROA}_{i,t} = \ & \beta_0 + \beta_1 \text{GOV}_{i,t} + \beta_3 \text{LNTA}_{i,t} + \beta_2 \text{PPE}_{i,t} + \beta_4 \text{CAPEX}_{i,t} + \beta_6 \text{GROWTH}_{i,t} + \beta_7 \text{CASH}_{i,t} \\
& + \beta_5 \text{LEVERAGE}_{i,t} + \text{IndustryFixedEffect}_t + \text{YearFixedEffect}_t + \varepsilon_{it}
\end{aligned}
\tag{5}
$$

The three corporate ESG performance disclosure elements results are presented in Table 7. The results show a strong relationship between a firm's environmental (ENV, column 2), social (SOC, column 3) and governance (GOV, column 4) elements and ROA for all firms at a statistical level of 1% significance. These results advocate the significant and favourable association between firms' ENV, SOC and GOV performance and ROA, which is consistent with prior literature and supports the stakeholder theory consistent by Nizam et al. [89]. The coefficient for ENV is positively correlated with ROA and statistically significant at the 1% level (*t*-statistic = 3.71).

Similar to the ENV, the SOC's coefficients show a positive relationship with ROA (*t*-statistic = 3.91). This is not consistent with prior studies that found that corporate social (SOC) performance negatively and significantly affects financial performance [90]. However, the latter study measured firm market performance by Tobin's Q, which may not reflect the actual financial performance of a firm because Tobin's Q estimates whether a given business or market is perceived to be overvalued or undervalued (i.e., the market value of a company divided by its assets' replacement cost). Also, Makni et al. [91] found no significant association between the six components of social performance (i.e., community and society, corporate governance, employees, customers, environment and human rights) and financial performance. The current study results are based on a composite index for SOC, which may explain the different results and reflect the overall impact of SOC performance disclosure. These findings reflect the lack of consensus over the past decade about the relationship between SOC and financial performance [21]. These scholars identified that the main cause of no consensus was the lack of uniform application of CSR and financial performance assessment methods with established linkages between CSR and financial performance. However, they did report that a positive (or neutral) relationship between CSR and financial results was observable in major studies. Therefore, the current SOC-ROA result is consistent with the majority of the prior studies.

Also, the coefficients for GOV show a favourable association with ROA (*t*-statistic = 5.29), which is in line with previous literature where firm governance is the primary force of a firm's sustainable behaviour [56,58].

With a robust negative coefficient, the firm's size (LNTA) shows the same results for all three ESG components for the other control variables. Conversely, there were favorable significant associations between not only all three ESG components and ROA but also growth (the firms' revenue growth) and cash (the firms' liquidity). In contrast, the PPE

(property, plant, and equipment) ratio is associated negatively with the capital expenditure ratio (capex), reporting a positive association, and the results show no significant association for all three ESG components. Also, there is a significant negative associated between leverage and ROA for all three ESG components. The other variables' findings for all disaggregated elements of ESG performance are reported consistently.

Results for the three elements of firms' disclosed ESG performance associations with either large or SME firms' financial performance, separately, are presented in Tables 8 and 9.

The three ESG and financial performance for large firms show significant results in Table 8, similar to the findings reported in Table 7. However, no significant associations are found (as presented in Table 9) for aggregated and the three ESG elements and SME firms' financial performance.

## 5. Robustness Tests

A number of tests were run to evaluate the findings' accuracy. The robustness test results, discussed below, support the main hypotheses.

We follow Fatemi, Glaum and Kaiser [31], Attig et al. [92], and El Ghoul et al. [93] in using the instrumental variable (IV) method to re-evaluate our main estimation models and then report the results. Considering that firms demonstrating better operational performance in the past appear to maintain a higher ESG disclosure score, the IV approach helps control any potential endogeneity bias initiated by reverse causality. It is also important to consider the impact on the results of unobserved firm-specific variables or missing elements [94]. This concern is properly addressed by including year and industry fixed effect, therefore including time-invariant unobservable heterogeneity. More than that, this study includes additional endogeneity analysis to address these issues. Following the previous literature by Cheng et al. [95], and Gupta and Krishnamurti [96], we performed a simultaneous equation model to find the appropriate instrument by using yearly means of firms' ESG performance disclosure as an instrument, recommended by Cheng, Ioannou and Serafeim [95]. This independent variable is likely to be exogenous to the firm's ESG performance level. The results are presented in Tables 10 and 11.

**Table 11.** Variance inflation factor (VIF) analysis.

| Variables | ROA | | | | | | | |
|---|---|---|---|---|---|---|---|---|
| | VIF | 1/VIF | VIF | 1/VIF | VIF | 1/VIF | VIF | 1/VIF |
| ESG | 2.0087 | 0.4978 | | | | | | |
| ENV | | | 1.7115 | 0.5843 | | | | |
| SOC | | | | | 1.2855 | 0.7779 | | |
| GOV | | | | | | | 1.6257 | 0.6151 |
| LNTA | 2.7287 | 0.3665 | 2.5551 | 0.3914 | 1.9733 | 0.5068 | 2.42 | 0.4132 |
| PPE | 2.2071 | 0.4531 | 2.402 | 0.4163 | 2.2694 | 0.4407 | 2.1882 | 0.457 |
| Capex | 2.0762 | 0.4816 | 2.1903 | 0.4566 | 2.0547 | 0.4867 | 1.9715 | 0.5072 |
| Growth | 1.0334 | 0.9677 | 1.0345 | 0.9667 | 1.026 | 0.9747 | 1.0336 | 0.9675 |
| Cash | 1.294 | 0.7728 | 1.2113 | 0.8256 | 1.2802 | 0.7811 | 1.2876 | 0.7772 |
| Leverage | 1.5163 | 0.6595 | 1.7566 | 0.5692 | 1.5825 | 0.6319 | 1.6022 | 0.6241 |
| Mean VIF value | 1.8378 | 0.5998 | 1.8374 | 0.6014 | 1.6388 | 0.6613 | 1.7325 | 0.6230 |

This table provides the results of the VIF test. The mean VIF value is the average value of the statistics.

In line with the main results, the robustness analysis (presented in Table 10) shows that ESG performance is favourably associated with ROA (*t*-statistic = 5.02). This shows that endogeneity does not drive our main findings. The robustness tests for the other disaggregated ESG performance elements follow the main results in our estimation models in Table 7.

Following control procedures by Brogi and Lagasio [84], and Kim et al. [97], we control for multicollinearity in the estimation models. Multicollinearity conducts the variance inflation factor (VIF) test and presents the results in Table 11. The VIF test spans values

from 1 and above. The higher the VIF value, the less trustworthy the estimation model is. Any VIF value higher than 10 is a sign of serious multicollinearity that needs further investigation [98]. The results of running the VIF test for the main models show values less than 3. This supports the fact that the estimation models in this study are far removed from having multicollinearity issues, confirming the reliability of the regression analysis.

## 6. Discussion

The aim of this study was to investigate whether firms' ESG performance is related to enhanced financial performance and whether this association differs between large and SME firms. This study sheds light on the importance of a firm's ESG performance in improving its sustainable behaviour globally by broadening scholarly understanding related to legitimacy theory. As noted earlier, the managerial branch of stakeholder theory suggests that poor ESG efforts of stakeholder groups will be influenced by threats to firms' ongoing survival [5], whereas strong ESG efforts predict increased company reputation, maintain accountability [39], and improve financial performance [9,99]. The reality is that ESG scores and their measures are also dependent on significant resources for providing ESG data [22].

A better understanding of the associations between three disaggregated elements of a firm's ESG performance (environmental, social and governance) and financial performance is achieved by expanding this panel data regression model analysis by the inclusion of years and industry effects. The inclusion of these effects in the estimation models invalidates any anxieties about the unobserved time-invariant or missing variables. Furthermore, alternative ESG performance disclosure measures were used and the results remain strong, which were confirmed after utilizing the instrumental variable (IV) approach.

The findings support a positive association between firms' aggregated ESG performance and profitability consistent with the first hypothesis. Of particular relevance was the strength of the results over ten years using panel data clearly illustrating the associa-tion between ESG and firm performance and perhaps bringing into question economic data collection at a single point in time. Therefore, the results may indicate that over shorter periods, e.g., one year –notwithstanding positive correlations of previous studies– the referent of ESG-performance associations are individuals' reflections as aggregated indicators only of the effects of ESG on performance, can be questioned. In the current study, however, all firms' ESG performance elements show a positive relationship with profitability for large firms but not SME firms. These findings are consistent with prior studies by Dalton, Daily, Johnson and Ellstrand [56] and later by Nollet, Filis and Mitrokostas [58] that firm's governance is the primary force of ESG performance. However, this paper's novel contribution to time-based data adds to this substantive literature about the importance of longitudinal data to measure ESG and firm performance. For instance, we broaden existing research by Klein and Zwergel [22], and Rabaya and Saleh [58] that ESG scores alone do not accurately reflect a firm's sustainability perfor-mance. Even while larger firms have greater resources to access ESG data, this may not be enough to accurately measure sustainable ESG performance. Rather, our findings suggest that the longitudinal nature of panel data is a far more accurate prediction of ESG performance criteria justifying the approach taken in this study. We contend that a better overall reflection of firms' ESG measures over time may in fact help those firms to achieve a sustainability competitive advantage.

The results for H2 show no significant association between any of the three components of ESG, either individually or collectively, for the overall ESG performance index and profitability, in relation to SME firms. This is inconsistent with the motivations explanation of both legitimacy and stakeholder theories and the underlying unwritten social contract. The main inconsistency with the results revolves around the reciprocal arrangement from gaining legitimacy or satisfying stakeholders' internal value chain (or supply chain) needs. Stakeholder theory argues that firms are motivated by the association between ESG and increased profitability because downstream customer stakeholders should impact the firms' profitability. However, in interpreting the results related to H2, the incremental increased

cost of gathering and disclosing the information may exceed the incremental increase in profitability. Given this study's very large sample size (*n* = 3422), aspects of legitimacy and stakeholder theory [5,6] can be questioned the perceived theoretical social contract between stakeholders and the community in relation to community ESG expectations. Therefore, the results of this study challenge claim through traditional legitimacy and stakeholder theory that ESG performance may be improved because of these associations, particularly among the SME data included in this study.

However, existing studies help to explain H2 results partially. Bianchi, et al. [28] reported that financial performance improved through a model simulation by using a dynamic performance management system (DPM) approach over eight years (2012–2020). DPM was a performance management system designed to pursue sustainable development in SMEs tailored to the characteristics of SMEs "that may be differentiated from large companies". However, DPM may be a cost that exceeds the available resources of many SME firms. Second, Rabaya and Saleh [59] found that integrated reporting (IR) had a mediating strengthening effect on the association between ESG and a firm's competitive advantage. They suggested that the IR format may help stakeholders understand the association between sustainability practices and the firm's increased performance and value.

Also, SMEs may not practice IR and therefore are not bound or motivated to the same extent to comply with ESG performance links, where prior research has found that only size was a significant positive predictor of social disclosure practices [100]. For instance, while corporations are said to lack an integrated analytical framework for creating indicators and indices for measuring enterprise sustainability [34,101], SME firms may be similarly situated plus lack the incentive that larger firms gain through regular use of audit committees and influence from a higher proportion of non-executive directors that help to increase ESG disclosure [102]. This is in addition to the environmental and social implications of non-compliance. Put simply, SMEs may lack the same level of stakeholder engagement and feedback from stakeholders to improve decision making and accountability such that external perceptions from society matter through a legitimacy theory lens [103]. Similarly, not only do SMEs lack resources for providing ESG data [22], but there is also a misunderstanding of sustainability leadership practices [104] required to implement sustainable ESG outcomes.

Also, the small customer base in a limited market and the reactionary characteristics of SMEs, described in prior literature (e.g., Hudson et al. [105], and Janang et al. [106]), may be the source of SMEs not practising IR and disclosure practices such that ESG disclosure may not have the same level of impact on the profitability of SME firms as previously thought.

### 6.1. Implications for Practice

This study has several implications for firms, market participants, other stakeholders, and regulators. Firms' primary and favourable implication is that improving ESG performance could enhance a firm's financial performance in the long run. Therefore, corporate ESG performance benefits shareholders and other stakeholders of large firms and creates a win-win situation. Managers should thus try to improve corporate ESG performance disclosure to foster sustainable profitability, and corporate governance should be integrated into long-term corporate strategies to sustain positive implications for financial performance. For SMEs, our study found that they do not receive financial performance benefits from ESG disclosure. Similarly, the findings of this study have practical implications for managers and stakeholders more generally. Improvement in the firm's ESG performance benefits financial performance and therefore is also beneficial for shareholders and other stakeholders in the long run [63,106]. Managers must target enhancing ESG performance to impact sustainability. Integrating firm governance into long-term strategies is more likely to enhance firms financially over time.

Regulators, furthermore, should continue to promote the responsible conduct of SMEs to enhance ESG awareness, particularly in relation to legitimacy and the sustainable expectations of society [34]. While the current study found no association between ESG and

performance for SMEs, this suggests that ESG-performance strategies are an opportunity for SME owners and managers. While resources, time and energy and other factors influence the ESG-performance link, regulators might result in future support for firms' ESG performance by adopting medium-sized firms' disclosure based on social and environmental standards. These regulations should be introduced to encourage firms to drive environmental awareness. It could also be extended by enhancing investment in firms with higher ESG performance, new sustainable products with better ESG-related features and higher stakeholder interaction, irrespective of size.

### 6.2. Theoretical Implications for Future Research

Future research can be extended to include unlisted firms or SMEs with diverse reputational viewpoints from those of large firms. Future research can also evaluate how different economic and market conditions influence a firm's ESG-performance association with profitability, such as those in emerging economies. However, consistent with our contributions to research, future research should consider taking advantage of the availability of data relating to, not just Australian firms but, any country's publicly listed data to derive a strong population for research. This is particularly relevant to longitudinal data. Future research may then be able to replicate the current study by including their own variables for panel regression analysis, thus helping to confirm the current research approach.

The proposed European Directive 2021/01014, that has been under discussion through seven discussion stages within the council of European Union since 20 April 2021 until 23 February 2022, highlights ESG's disclosure importance for European-based firms, including SMEs.

Riva et al. [107] concluded that the communication of non-financial information by SMEs has a dual objective. The disclosure demonstrated, firstly, the existence of a corporate governance structure and, secondly, represented a management and accounting tool had been designed to function adequately to ensure a firm operates as a going concern. They extend their argument with the comment that it is essential that firms know how to communicate with stakeholders, which may be clearly illustrated through financial and non-financial data in the form of a dashboard. These dashboards provide readers with not only an indication of whether the company is healthy, or otherwise, but also whether it is capable of coping with critical macro and microeconomic issues that may have arisen as a result of a pandemic, which is an exogenous factor to the firm. Future research accordingly should include an investigation into the motivations of ESG reporting by SME firms, especially during this period of ongoing uncertainty caused by COVID on profitability.

### 6.3. Limitations of the Study

There are two limitations. First, while several firms' financial characteristics (such as LNTA, PPE, cash, and leverage) are contained within this study's analysis, other moderating characteristics are not included. For example, different ownership structures, the presence of an ESG committee, or the competition in the market can potentially impact firms' ESG performance. Second, the sample involves only publicly listed firms, which hinders the generalizability of the results.

## 7. Conclusions

This study examined the ESG performance of large and SME firms, finding that ESG performance has a positive relationship with profit in large firms but not in SME firms. The authors found that while some of these findings are consistent with prior studies as discussed, SME firms' priorities for collecting ESG data differ significantly compared to large firms' aspirational and environmental needs. For instance, we noted throughout that the large firms are motivated by stakeholder and other needs, while SME firms do not have the same priorities. Similarly, small and nascent firms may not have the resources of large firms, suggesting that competitive factors related to downstream networking will

markedly differ for both groups of firms. Moreover, this study highlights the value of using longitudinal data and the results help to broaden existing research related to interpreting the ESG initiatives and priorities of firms more generally.

**Author Contributions:** A.G. contribution was the conceptualization, methodology and formal analysis, and writing original draft. P.A.M. contribution was writing original draft and review and editing. J.S. contribution was the conceptualization, methodological design, writing original draft, and review and editing. All authors have read and agreed to the published version of the manuscript.

**Funding:** This research received no external funding.

**Institutional Review Board Statement:** Not applicable.

**Informed Consent Statement:** Not applicable.

**Data Availability Statement:** The secondary source of the data examined was extracted from Bloomberg's database.

**Conflicts of Interest:** The authors declare no conflict of interest.

**Appendix A**

**Table A1.** Variable Definitions.

| Category | Measure | Definition/Measurement |
|---|---|---|
| Environmental, Social and Governance score | *ESG* | Calculated based on 120 indicators, including three elements of environmental, social and governance |
| | *ENV* | Environmental scores include GHG emissions, water, energy, biodiversity, products & services, and compliance |
| | *SOC* | Social score includes labour engagement and related decent work, society, human rights and product responsibility |
| | *GOV* | The governance score includes over-boarding and executive compensation |
| Firm profitability | *ROA* | An indicator of firm profitability percentage named Return on Assetscalculated based on the earnings before interest and tax (EBIT) divided by total assets earnings before interest and taxes (TA) $\text{ROA} = \frac{\text{EBIT}}{\text{TA}}$ |
| Firm characteristics: Firm size | *LNTA* | Natural logarithm of total assets |
| Leverage | *Leverage* | Leverage or debt ratio measured as total debts divided by total assets |
| Property, plant, and equipment | *PPE* | Property, plant, and equipment to total sales |
| Capital expenditure | *Capex* | Capital expenditure divided by total sales |
| Revenue growth | *Growth* | Percentage change in sales over the prior year |
| Cash | *Cash* | Cash divided by total assets |

Source: Bloomberg database.

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
