# Peer review of "Environmental, Social, Governance & Financial Performance Disclosure for Large Firms: Is This Different for SME Firms?"

_sustainability, doi:10.3390/su14106019_

Round 1

Reviewer 1 Report

The article is interesting, and the researched problem has scientific potential. However, some problems need to be solved:

 1. Data processing is performed using descriptive statistics. The article would gain value if complex statistical methods were used to establish the relationships among variables (SEM) in the context of content analysis.

 2. I think more theoretical implications would raise the quality of the paper.

In addition, the authors need to describe more clearly the steps of their research process (eventually into graphic illustration) into the Methodology section. 
Also, they have to describe the methods used in other similar research and to explain why their method brings better results (findings) than other research. 

The paper has scientific value and can be published after carefully reviewing the reported issues.

Reviewer 2 Report

1. This study analyzed the relationship between ESG and financial performance for Australian companies. There is a lack of explanation as to why the subject was validated against Australian companies. In addition, it is necessary to explain what characteristics or differences Australian companies have from other continental companies.

2. A number of analyses have been conducted on the relationship between EGS and financial performance according to the size of the company. It is necessary to mention the differences from prior studies conducted on similar topics and clearly present the contribution of this study compared to previous studies. 

3. ROA was used as a measure of financial performance. Recent previous studies prefer TOBIN's Q as a measure of financial performance. It is necessary to present the verification using TOBIN's Q as an additional analysis. 

4. Please present the number of large firms and SMEs by year in table 1.

5. In the Data and Methodology section, please mention a clear definition of SME enterprises. Also, I would like you to clarify how to measure SMEs. 

6. In the Estimation model section, Model (1) and Model (2) are identical. This creates confusion for readers. It is necessary to clarify the contents of the paper by changing the name of some variables.

7. The conclusion overlaps a lot with the introduction and the content of the text. It is necessary to reduce duplicates and present the conclusion more concisely.

I hope that the reviewer's comments will help the constructive development of the paper.

Round 2

Reviewer 1 Report

Dear authors,

Thank you for making the improvements to your paper.

I believe that your work can be published in this form.

Reviewer 2 Report

Several things I pointed out and the requests for correction were properly reflected.

The authors have made great efforts to meet the requirements.

I believe it is a paper that can be published, and I hope that the authors can actively carry out follow-up research.

This manuscript is a resubmission of an earlier submission. The following is a list of the peer review reports and author responses from that submission.